# Flexibility of Boolean Network Reservoir Computers in Approximating Arbitrary Recursive and Non-Recursive Binary Filters

**DOI:** 10.3390/e20120954

**Published:** 2018-12-11

**Authors:** Moriah Echlin, Boris Aguilar, Max Notarangelo, David L. Gibbs, Ilya Shmulevich

**Affiliations:** 1Institute for Systems Biology, 401 Terry Ave N, Seattle, WA 98109, USA; 2Molecular & Cellular Biology Program, University of Washington, Seattle, WA 98195, USA

**Keywords:** complex adaptive systems, systems dynamics, dynamical systems, signal processing, reservoir computing, machine learning, Boolean networks, biological modeling

## Abstract

Reservoir computers (RCs) are biology-inspired computational frameworks for signal processing that are typically implemented using recurrent neural networks. Recent work has shown that Boolean networks (BN) can also be used as reservoirs. We analyze the performance of BN RCs, measuring their flexibility and identifying the factors that determine the effective approximation of Boolean functions applied in a sliding-window fashion over a binary signal, both non-recursively and recursively. We train and test BN RCs of different sizes, signal connectivity, and in-degree to approximate three-bit, five-bit, and three-bit recursive binary functions, respectively. We analyze how BN RC parameters and function average sensitivity, which is a measure of function smoothness, affect approximation accuracy as well as the spread of accuracies for a single reservoir. We found that approximation accuracy and reservoir flexibility are highly dependent on RC parameters. Overall, our results indicate that not all reservoirs are equally flexible, and RC instantiation and training can be more efficient if this is taken into account. The optimum range of RC parameters opens up an angle of exploration for understanding how biological systems might be tuned to balance system restraints with processing capacity.

## 1. Introduction

Many biological systems are regarded as non-linear dynamical systems [1] operating in high-dimensional spaces. Proteins, genes, and macromolecules interact in a variety of ways to create the dynamics of cells [2]. Collections of cells interact and coordinate activity, forming cohesive units such as bacterial colonies, simple multicellular organisms, or tissues in more complex multicellular organisms. At each level of organization, ‘input’ signals (e.g., odors or hormones) are introduced into the system, processed by means of the system’s dynamics, and responded to accordingly, sometimes generating new ‘output’ signals as a by-product [3]. Fundamentally, biological systems must process external signals in real time to inform a wide variety of response decisions. For example, the fruit fly olfactory system projects an input to a high-dimensional space before classifying an odor [4].

In this vein, reservoir computing (RC) is a form of signal processing that is used for classification and online learning and uses such high-dimensional systems to process signals. Initially called echo state networks [2] or liquid state machines [5], the computational structure resembles an unorganized recurrent neural network. These networks were inspired by the most classical example of signal processing in biological systems: the brain. While initial research in neural networks was focused on modeling and understanding neural computation [6], applications and directions for research quickly expanded in scope. For reservoir computers, this includes systems for predicting time series from chaotic systems [7,8] as well as multiple real-world applications such as gas detection [9], robotics [10,11,12], and economic trends [13]. Applications in processing human data are also found with human emotion [14] and gesture recognition [15], speech and text processing [16,17], and health care monitoring [18].

In biology, reservoir computers have been used to analyze biological data for identifying anomalous states such as cardiac arrhythmia [19], seizures [20,21], and microsleeps [22], as well as for high-dimensional classification such as cell type detection [23]. In these applications, RCs are used to interpret or classify biological data. RCs have also been used as models of biological systems, serving to explore the mechanisms and dynamics of said systems. For example, the questions of how the brain functions [24,25,26] and how gene regulatory networks respond to external stimuli [27,28] have been studied in an RC context.

In reservoir computing, an incoming signal is fed into the reservoir, which is a randomly connected recurrent network where nodes are connected via a variety of coupling functions. Thus, the reservoir transforms the signal into a high-dimensional representation. Finally, an output layer is trained, often with straightforward techniques such as regularized regression [29], in order to perform a classification or regression task (Figure 1).

RCs can also be trained to act as signal processors, filtering the input to produce a new output through approximating a function that is normally applied to a window sliding over the signal. The type of function can be extended to recursive functions, where some elements of the input window are replaced by values from previously generated outputs. These types of filters have a long history of use, which includes filtering biological signals [30,31,32].

While reservoirs are most typically built using unorganized recurrent neural networks, it has been shown that any non-linear dynamical system that exhibits a fading memory property can theoretically be used to compute a time-invariant function on a signal [33]. In fact, DNA, carbon nanotubes, memristor networks, and even buckets of water have been shown to work as reservoirs [34,35,36]. One type of dynamical system that has only recently been explored as a reservoir is Boolean networks [37]. Boolean networks (BNs) are well-suited to reservoir computing, since they are one of the simplest modeling approaches that (1) can capture the heterogeneity, both in wiring and coupling functions, that characterizes self-assembled systems; and (2) can exhibit the non-trivial dynamical behavior that is required for computation [38]. Additionally, the fading memory property can easily be achieved in Boolean networks by adjusting two key parameters: the average in-degree of the Boolean functions, $\bar{K}$, and the bias, *p*, which is the probability that a function produces an output of one for any given set of inputs [39,40]. The implementation of Boolean networks, both in software and in hardware, is also considerably easier and faster than more traditional approaches [41].

Similar to neural networks, Boolean networks were first used as models of biological systems, namely gene regulatory networks [37,42,43]. While often researched to understand cellular behavior in terms of protein/gene interaction networks in isolation [44,45,46], BNs have also been used to understand how external signals impact biological systems [47,48,49]. These systems rely on their ability to process external signals (relevant to survival or population function) and make decisions based on them in real time, thereby performing a similar computation to a reservoir [4]. Just as the Boolean network framework has provided insight into how biological systems function, studying BN reservoir computers (BN RCs) may further our understanding of biological signal-response.

Previous work on BNs as reservoirs is still limited. Snyder et al. [38,50] evaluated BN RCs with different numbers of nodes (N) and average node input degrees ($\bar{K}$). Using a measure of reservoir quality, it was found that compared to networks with homogeneous in-degrees, networks with heterogeneous in-degree better separated inputs by class: the so-called separability property [5]. Also, it was found that computational ability—when the difference in separability and fading memory is the greatest—was maximized when the average in-degree was $\bar{K}$ = 2 (out of $\bar{K}$ = 1,2,3) [50], and that higher $\bar{K}$ values led to worse performance. This value for $\bar{K}$ is notable, as Boolean networks with $\bar{K}$ = 2 are known to be dynamically critical (Lyapunov exponent = 0). Critical dynamics, which lie at the border between order and chaos, have been shown to be important in computation and information processing, biological and otherwise [51,52,53,54,55].

While previous work [38,50] has laid the foundation showing that it is possible to use Boolean networks as reservoirs, the main focus has been on testing a small number of well-known functions, such as the median and parity functions, as benchmark measures. We are extending the body of research in three ways: (1) by performing a wider examination of how well these systems perform across different functions, including recursively defined operators; (2) by analyzing how flexible a single reservoir is for repurposing (i.e., retraining only the output layer and keeping the same network reservoir); and (3) characterizing the functions that are easier or harder for the reservoir to implement in terms of the function’s average sensitivity, which is a measure of its smoothness. We also evaluate reservoirs under different dynamical regimes: ordered, critical, and chaotic.

## 2. Materials and Methods

### 2.1. Reservoir Computer

A reservoir computer consists of three components: the input layer, the reservoir network, and the output layer (Figure 1). The input represents a temporally changing signal that perturbs the reservoir, which performs computations on the signal in real time. The output node is the readout of the reservoir’s computation, estimating a given function operating on the signal. The value of the output node is given by a linear combination of the states of the reservoir nodes that are continuously being driven, either directly or indirectly (via other nodes), by the input signal. The weights of the linear regression are trained to be specific to a given objective function, which captures the error between the reservoir output and the function to be approximated.

### 2.2. Reservoir

In our implementation, the reservoir is constructed as a random Boolean network (RBN) [37]. RBNs are networks with N binary-valued nodes with states:$$X_{t}=\left\{{x_{1}}^{t},\;{x_{2}}^{t},\;\ldots ,\;{x_{N}}^{t}\right\}\;{x^{t}}_{i}\in \left\{0,1\right\},\;i=1,\ldots ,N,\;t\;>\;0$$

The state of the network at any time, $X_{t}$, is a function of the state at the previous time, $X_{t-1}$, given by:$$X_{t}=F\left(X_{t-1}\right)$$
where:$$F=\left\{f_{1},\;f_{2},\;\ldots ,\;f_{N}\right\}$$
is the set of Boolean updating functions corresponding to each node. The node in-degree does not need to be constant, such that each function $f_{i}$ has an independent number of arguments, $k_{i}$, so that:$${x_{i}}^{t}=f_{i}\left({x_{j_{1}}}^{t-1},{x_{j_{2}}}^{t-1},\;\ldots ,\;{x_{j_{k_{i}}}}^{t-1}\right)$$

The identity of each of the $k_{i}$ arguments is chosen randomly from the N nodes, without replacement. The in-degree of the whole network, $\bar{K}$, is characterized by the average in-degree across its nodes. The other primary descriptor of the network is the bias, p, which is the probability that a function outputs a one. Together, p and $\bar{K}$ can be varied to adjust the dynamics of the network, ranging from ordered, in which perturbations die out, to chaotic, in which perturbations are amplified [39,40]. For this paper, we will fix p = 0.5 and tune the dynamics of the networks by varying $\bar{K}$. Generally, we tune the dynamics using $\bar{K}$ < 2 for ordered, $\bar{K}$ = 2 for near critical, $\bar{K}$ > 2 for chaotic. To construct a network with a specific $\bar{K}$, we take $\bar{K}\cdot N$ edges uniformly distributed amongst all of the pairs of nodes.

### 2.3. Input

In order for the RBN to act as a reservoir computer, it must be driven by an external signal, which is represented by a temporal sequence of binary values, $u^{t}$. Some percentage (expressed as a fraction) of reservoir nodes, L, is directly connected to the input signal. If a node, i, is connected to the input signal, then the input becomes an additional argument to its function, i.e.:$${x_{i}}^{t}=f_{i}\left({x_{j_{1}}}^{t-1},{x_{j_{2}}}^{t-1},\;\ldots ,\;{x_{j_{k_{i}}}}^{t-1},\;u^{t-1}\right)$$

The L·N nodes that have the input as an additional argument are chosen uniformly from the N nodes of the reservoir without replacement. 

### 2.4. Output

At each time step, the reservoir produces a binary output value, ${\hat{y}}^{t}$, which is defined as: $${\hat{y}}^{t}=\Theta \left(\sum_{j=1}^{N}w_{j}x_{j}^{t}+b\right)$$
where Θ is the function that maps values greater than 0.5 to 1, and everything else to 0; $w_{j}$ is a weight for each node of the reservoir, and b is a constant. The parameters $w_{j}$ and b are trained to approximate the output, $y^{t}$, of a given Boolean function operating on a moving window over $u^{t}$, under some error criterion. Here, we use every node in the reservoir in the linear regression so that an optimal combination may be found. Pre-specifying which nodes are used in calculating the output would limit the success of the reservoir, since the structure and rules of the reservoir are fixed. In practice, when N = 100 and L = 0.5, ~25% of the nodes have non-zero weights, which are distributed equally across the nodes that are directly and indirectly perturbed by the signal.

### 2.5. Objective Functions

In this work, the computational performance of the reservoir computer is assessed by approximating the Boolean functions of three and five arguments evaluated over a temporally changing input signal. Moreover, we considered two types of functions:Non-recursive functions, defined as $y^{t}=g\left(u^{t-\tau },u^{t-\tau -1},\;\ldots ,u^{t-\tau -\left(M-1\right)}\right)$Recursive functions, defined as $y^{t}=g\left(u^{t-\tau },u^{t-\tau -1},\;\ldots ,u^{t-\tau -\left(M-2\right)},\;y^{t-1}\right)$
where M is the length of the sliding window on the input signal for which a function is approximated, and τ is a delay between when the signal perturbs the reservoir and when the corresponding output is computed. In this work, we explore M = 3, 5 for non-recursive functions, and M = 3 for recursive functions. Here, τ is kept fixed to τ = 1. We have tested all 256 three-bit recursive and non-recursive functions, and 1000 randomly sampled five-bit functions.

### 2.6. Training and Testing Algorithm

To train a BN RC for approximating a single function, we use random binary sequences as input streams, $u^{t}$, and compare the reservoir’s output value, ${\hat{y}}^{t}$, to the function output, $y^{t}$, evaluated over the input stream. Specifically, we use a set of 150 random binary sequences of length 10+M−1 (e.g., 12 for 3-bit functions) to generate a set of P = 1500 different $\left(\hat{y},\;y\right)$ pairs. It should be noted that for each 10+M−1 binary sequence, the reservoir is randomly initialized, and no transient period occurs before the outputs are used for training. With the $P\;\left(\hat{y},y\right)$ pairs, a regression model is fit in order to generate the coefficient weights on the output layer. We used Scikit-learn to perform lasso regression [56] with an α = 0.1 for fitting.

To test a BN RC for approximating a single function, we again use a set of 150 random binary sequences to generate $P\;\left(\hat{y},y\right)$ pairs. We then measure the accuracy of the BN RC for that function, which is defined as:$$a_{ij}=1-\rho /P,$$
where j indexes the Boolean function, i is a particular instantiation of an RC, and ρ is the summed error $\sum_{l=1}^{P}\left|{\hat{y}}^{l}-y^{l}\right|$, which is scaled by *P*. Thus, it can be seen that the accuracy $a_{ij}$ is between 0–1. This process of training and testing is repeated for a set of different functions for each BN RC, thereby creating multiple estimation accuracies $a_{ij}$.

### 2.7. Overall Strategy

The goals of this work centers around exploring how the approximation accuracy of RBN RCs varies across many different functions, providing a sense of ‘flexibility’. To do that, one reservoir is constructed and applied to estimating the output of a set of Boolean functions, such as all of the three-bit functions. This is done for a total of 100 RC instances for each combination of different values of N, L, and $\bar{K}$ to test the effect of reservoir size, degree of network perturbation by the signal, and dynamical regime, respectively. We use N = 10, 20, …, 50, 100, 200, …, 500; L = 0.1, 0.2, …, 1; and $\bar{K}$ = 1, 2, 3. For example, with the three-bit functions, we trained three (values of $\bar{K}$) × 10 (values of N) × 10 (values of L) × 100 (RCs instances) × 256 (3-bit functions) = 7,680,000 RBN RCs.

Each RC is trained and tested for approximating a set of functions, including median and parity, reporting an accuracy for each function. We chose three sets of functions for testing the RCs. First, there are the three-bit functions, for which we test all possible $2^{2^{3}}=256$ functions. Second, there are five-bit functions, for which there are $2^{2^{5}}=4,294,967,296$ functions, which is a much larger function space than that for three bits. Working with the complete function set is impractical, so we made a random sample over the function space, choosing to test 1000 functions. Some additional hand-selected key functions, such as median and parity, were also used.

The last set of functions was recursive three-bit Boolean functions. This set of functions is defined as taking a set of arguments that includes the last produced output, $y^{t}=g\left(u^{t-\tau },\;u^{t-\tau -1},\;y^{t-1}\right)$. This is a more difficult task, because the BN RC must approximate the function with a hidden variable, $y^{t-1}$.

## 3. Results

### 3.1. Benchmark Functions: Median and Parity

We start our analysis with the approximation of the two functions tested in previous BN RC works [38,50]: the temporal median and parity functions. The median function calculates whether there are more 1 s than 0 s in the window, and is defined as:$$d_{t}=\chi [\sum_{i=0}^{M-1}u\left(t-\tau -i\right)>M/2]$$
where χ[A] is an indicator function that gives a one if A is true, and zero if A is false. The parity function calculates whether there is an odd number of ones in a bit string. It is defined as:$$p_{t}={⊕}_{i=0}^{M-1}u\left(t-\tau -i\right)$$
where ⊕ is an additional modulo 2. These two functions are frequently used as benchmarks for reservoir performance. For each function $g_{j}$, we trained 100 BN RCs for each set of the parameters $\bar{K}$, N, and L, measuring the mean accuracy with respect to the 100 BN RCs:$${\bar{a}}_{j}=\frac{1}{100}\sum_{i=1}^{100}a_{ij}$$

### 3.2. Median

On average, the three-bit median function is approximated with high accuracy (Appendix A). All of the BN RCs perform better than random, with the lowest mean accuracy of ~0.7 for N = 10 and L = 0.1. Increasing N and L increases the accuracy logarithmically up to ~0.98 for N = 500 and L = 1 (Figure 2A). However, reservoirs of size N > 200 already have near-perfect performance (>0.9), and increasing L has a minimal effect on these reservoirs. The general trend in the accuracy in relation to N and L remains the same for the five-bit median (Appendix A). However, the overall accuracy is lower relative to the three-bit median (Figure 3A). 

The average in-degree, $\bar{K}$, of the reservoirs also affects the reservoir performance. For the three-bit case, the performance is clearly reduced for $\bar{K}$ = 3, with those reservoirs having the lowest accuracy. $\bar{K}$ = one and two have similarly high levels of accuracy (Figure 2A). However, increasing the size of the function window from three to five creates a more obvious separation between these $\bar{K}$ values (Figure 3A). Interestingly, for most of the values of N and L, the reservoirs where $\bar{K}$ = 2 perform the best; however, for higher values of N and L, the accuracy for the reservoirs where $\bar{K}$ = 1 are the highest. Overall, these results qualitatively agree with previous work by Snyder et al. [38,50].

Unexpectedly, when the reservoir is tasked with approximating a three-bit recursive median, the performance is nearly as high as for the non-recursive three-bit median, and higher than that of the five-bit (non-recursive) median (Figure 4A). Two notable differences for the recursive median are that (1) increasing L does not grant as sharp of an increase in accuracy, and (2) the performance of $\bar{K}$ = 3 is more reduced as compared to lower $\bar{K}$ (Appendix A).

### 3.3. Parity

The accuracy curves for approximating the three-bit parity function have a strikingly different appearance than those for the median (Figure 2B). Accuracy is highly dependent on the value of L, with no reservoir performing better than random (~0.5) for L = 0.1. Very small reservoirs (N < 40) are incapable of high accuracy (Appendix A). As reservoirs become larger, the maximum accuracy increases. When the reservoirs are used to approximate the five-bit parity function, the performance drops significantly (Figure 3B). No reservoir is capable of performing better than random at L < 0.5, and an accuracy of >0.8 is only achievable with larger networks (N > 300) and L near 1 (Appendix A). These observations are consistent with previous work [38,50].

There is not much separation among the approximation accuracy curves by values of $\bar{K}$ of the three-bit parity, although $\bar{K}$ = three shows slightly lower performance on the task (Figure 2B). For the five-bit parity, reservoirs with different $\bar{K}$ values show a clear difference in accuracy (Figure 3B), with $\bar{K}$ = 2 reservoirs being best. As with approximating the median function, the accuracy of $\bar{K}$ = 1 reservoirs improves rapidly with L, changing from the lowest to the highest accuracy. The relative relationship between the accuracies of different $\bar{K}$-valued reservoirs that we observed is consistent with the above-mentioned previous work, but there are some discrepancies in the actual accuracies observed. We found a much better performance of $\bar{K}$ = 1 and 3 for the three-bit parity, and $\bar{K}$ = 1 and 2 for the five-bit parity. These differences are likely due to possible differences in how the average in-degree of the reservoir is computed. In this work, the L×N edges connecting the input to the reservoir are not included in calculating $\bar{K}$, whereas they are included in the work by Snyder et al.

The recursive three-bit parity appears to be extremely difficult to approximate. All of the reservoirs have an accuracy ~0.5, regardless of N and L, although there is an upward trend in accuracy associated with L. Interestingly, there is no evident improvement with increasing N values, and it is unclear whether reservoirs with more than 500 nodes would perform any better. Given the nature of the recursive parity function, it seems that a reservoir would need to “remember” the very first bit of the input signal, which may be too difficult for any reservoir to accomplish.

### 3.4. Estimating a Range of Functions

To assess the flexibility of reservoir topologies, all of the possible three-bit functions were tested. For each N, L pair, we measured the mean accuracy with respect to all of the functions and all of the reservoirs—e.g., $\bar{a}=\frac{1}{100}\frac{1}{256}\sum_{i=1}^{100}\sum_{j=1}^{256}a_{ij}$ for three-bit functions. Similar to what we observed with median and parity functions, $\bar{K}$ = 3 reservoirs showed lower performance on the task (Figure 5).

The observations made in the three-bit space were clarified in the five-bit function space. A sample of five-bit functions showed similar trends as seen in the three-bit function set, but with greater separation between dynamical regimes (Figure 6). However, the ability to approximate functions is noticeably lower compared to three-bit functions. At N = 100 and L = 0.5, all of the prediction accuracies are less than 0.75, regardless of $\bar{K}$. Compared to the performance in the three-bit function space where accuracy was above 0.95 using similar parameters, in the five-bit function space, it is not until the parameters are maximized that an accuracy of 0.95 is achieved.

Looking at the effect of $\bar{K}$ in the five-bit function space, the accuracy is increased with N and L for all of the $\bar{K}$ values, but $\bar{K}$ = 2 is most often the most accurate. The accuracy for $\bar{K}$ = 1 is higher than $\bar{K}$ = 2 after certain values of L. The value of L at which $\bar{K}$ = 1 reservoirs outperform $\bar{K}$ = 2 decreases as N increases. This effect is also partially seen in the three-bit functions (Figure 5) and three-bit recursive functions (Figure 7), although the relationship is less clear, except when looking at specific functions, such as the median and parity (Figure 3).

In terms of recursive functions, the accuracy is lower than both the non-recursive three-bit and five-bit functions. For N = 500, L = 1, and $\bar{K}$ = 2 or less, the accuracy is above 90%. It does not appear that accuracy above ~95% is possible with the current system. This is likely due to certain functions that are essentially impossible to approximate. This is discussed below, where the temporal order of the arguments to Boolean functions is shown to have an effect.

### 3.5. Reservoir Flexibility

One goal of this work was to assess the flexibility of reservoirs. A highly flexible reservoir can approximate a high number of functions without any changes to the network topology or size. To assess reservoir flexibility, we compute a flexibility metric $\Phi_{i}$ for each reservoir i, which is defined as:$$\Phi_{i}=median\left(a_{ij}-0.5\right)/\left(mad\left(a_{ij}\right)+1\right)$$
where the median and mad (median absolute deviation from the median) are taken over all of the functions $g_{j}$. Values of $\Phi_{i}$ are within [0, 0.5] and cannot exceed $median\left(a_{ij}-0.5\right)$. A reservoir with a low $\Phi_{i}$ value cannot approximate many (or any) of the functions well. As the number of functions that a reservoir can approximate increases, so does $\Phi_{i}$.

For each combination of N and L, we have a distribution of $\Phi_{i}$ values, one for each of 100 BN RCs. These can be visualized as density curves (Figure 8). With increasing values of N and L, the distributions become increasingly right skewed and sharply peaked; meanwhile, low N, L values are characterized mostly by reservoirs with low Φ values, and high N, L values are characterized mostly by reservoirs with high Φ values. However, for many intermediary values of N and L, there is a rather flat distribution of Φ values, indicating that there is heterogeneity of flexibility in the performance for reservoirs with these parameters (i.e., some are highly flexible, while some are not). The gradation of distribution shape with increasing N and L values depends on the value of $\bar{K}$ (Appendix A), although the overall trend remains. With higher values of $\bar{K}$, the distribution only shifts toward higher accuracies with higher values of L, N.

The flexibility of reservoirs approximating five-bit functions is markedly reduced compared to three-bit functions (Appendix A). For all but the highest values of N and L, most of the reservoirs have only moderate flexibility (Φ = 0.25). Unlike for three-bit functions, the shape of the distributions remains the same across N and L. Consistent with the effect of $\bar{K}$ on mean accuracy, the Φ distributions for $\bar{K}$ = 1 and $\bar{K}$ = 3 are shifted to lower values as compared to $\bar{K}$ = 2.

Surprisingly, the distributions of Φ values for reservoirs approximating recursive three-bit functions behave differently than those of non-recursive three-bit functions. This was unexpected, since the mean accuracy curves (Figure 7) for the recursive functions resemble those for the non-recursive functions (Figure 5). While Φ increases with N and L as it does for non-recursive three-bit functions, the change in the distributions does not follow the same pattern (Appendix A). For low N, L values, the distribution of Φ values is wide, and becomes increasingly narrow and peaked with increasing N and L values, as compared to the narrow peaky edges and flat wide middle of the non-recursive three-bit Φ distributions.

### 3.6. Determinants of Difficulty

To better understand why reservoirs do not perform uniformly well for all of the Boolean functions of a given size, we investigated the possible factors related to the functions to be approximated that could contribute to lower estimation accuracy. One possible way to compare Boolean functions with the same number of input variables is via the average sensitivity [57,58] of the function, ${\bar{s}}_{g}$. As the name suggests, this metric evaluates how sensitive the output of a function is to any change in the inputs. The average sensitivity of a function g is given by:$${\bar{s}}_{g}=E\left[s_{g}\left(\left[u^{1},\;u^{2},\;\ldots ,\;u^{M}\right]\right)\right]$$
where:$$s_{g}\left(\left[u^{1},\;u^{2},\;\ldots ,\;u^{M}\right]\right)=\sum_{i=1}^{M}\chi \left[g\left(\left[u^{1},\;u^{2},\;\ldots ,\;u^{M}\right]⊕e_{i}\right)\neq g\left(\left[u^{1},\;u^{2},\ldots ,\;u^{M}\right]\right)\right]$$
and $e_{i}$ is the unit vector with a one in the ith position and zeroes elsewhere, and χ[A] is an indicator function that gives a one if A is true and zero if A is false. The expectation is typically taken with respect to a uniform distribution over the M-dimensional hypercube.

A function that is insensitive has a low average sensitivity. For example, the constant function, $g\left(\left[u^{1},\;u^{2},\;u^{3}\right]\right)=1$, is not dependent on any of its inputs, and thus, ${\bar{s}}_{g}$ = 0. On the other hand, the three-bit parity, $g\left(\left[u^{1},\;u^{2},\;u^{3}\right]\right)=u^{1}⊕u^{2}⊕u^{3}$, has a high average sensitivity, as it will take a different value if any of its input variables are toggled, making ${\bar{s}}_{g}$ = M = 3 the maximum possible value. The sensitivity can be interpreted as the smoothness of a function. We hypothesized that a function with high sensitivity would be more difficult to estimate, because it requires more information about the inputs, and is harder to generalize.

To investigate whether there is a relationship between the average sensitivity and accuracy, we computed the average accuracy over all of the reservoirs, approximating all of the functions with a given average sensitivity, $\bar{a_{s}}$ (Figure 9 and Figure 10). Here, we only discuss the results for $\bar{K}$ = 2, since the results for $\bar{K}$ = 1,3 are comparable (Appendix A). For three-bit and five-bit functions, there is a clear inverse linear relationship between ${\bar{s}}_{g}$ and accuracy. As the average accuracy increases with N and L, the slope of the line becomes less steep, but the relationship remains. Additionally, the approximation of functions with greater ${\bar{s}}_{g}$ does not improve as much with larger reservoirs. For low N values, the relationship between sensitivity and accuracy for recursive functions closely matches that for the non-recursive functions. However, as N increases, the reservoir performance for recursive functions with high sensitivity remains relatively unimproved.

While the relationship between the average sensitivity and accuracy appears to be mostly linear, there are clear points of deviation from linearity, most notably at ${\bar{s}}_{g}$ = 1, 2, 3, 4. Functions with the same average sensitivities can have different degrees of dependence on each variable. The effect of each variable, $u^{i}$, on the output can be measured by the activity, $\alpha_{g}^{i}$, of that variable in function g [57,59]. The activity of a variable is the probability (hence, $\alpha_{g}^{i}$ is between zero and one) that toggling the variable’s value changes the output of the function. If a uniform distribution of input states is assumed, the sum of the activities is equal to the average sensitivity.

We examined the distribution of activities for three-bit functions, and found that the points of deviation (Figure 9) correspond to groups of functions with the same sensitivity, but different possible activities, e.g., ${\bar{s}}_{g}=1$, $A_{g}=\left[0,\;0.5,\;0.5\right]$ or $A_{g}=\left[1,\;0,\;0\right]$, where $A_{g}=\left[\alpha_{g}^{1},\alpha_{g}^{2},\alpha_{g}^{3}\right]$ (Figure 11A). For such functions, reservoir accuracy generally decreases when the variables that are observed farthest in the past have higher activities. Notably, when $\alpha_{g}^{2}$ and/or $\alpha_{g}^{3}$ = 1 (${\bar{s}}_{g}$ = 1,2), the function accuracy is significantly lower than that for other functions with the same ${\bar{s}}_{g}$ values. As a result, there are large dips in the mean accuracy at integer values of ${\bar{s}}_{g}$ (Figure 9). Thus, the performance of the reservoir is not only dependent on the sensitivity of the function, but also on the temporal distribution of activities (Figure 11B). For five-bit functions, the relationship between accuracy and activities is not as well defined, although it is still identifiable (data not shown).

Recursive functions behave similarly to non-recursive functions with some exceptions (Appendix A). Since the recursive variable is the most temporally distant, this variable having high activity causes an even greater reduction in accuracy. In fact, reservoirs cannot approximate any recursive functions with $\alpha_{g}^{3}$ = 1. There are also some cases in which reservoirs approximating functions with high activity on the recursive variable have higher accuracy compared to others with the same sensitivity. Future research could investigate whether this is related to the recursive nature of the function.

## 4. Discussion

In this work, we found that the flexibility of a particular BN RC instantiation is a function of the topology and size of the reservoir, as encoded by the parameter set ($N,L,\bar{K}$). Generally, higher values of N and L result in more flexible reservoirs; however, there is heterogeneity when N or L are low, and flexible reservoirs can be found at these values. As noted in research concerning BN RCs, the optimal parameter set includes tuning the dynamics toward criticality, which leads to more flexible systems. For signal processing, a flexible reservoir will be more accurate and more efficient, requiring less searching and training when different filters are being applied to the same data. For biological systems, where survival is dependent on signal response, N,L, and $\bar{K}$ can be tuned to balance the restraints of the system with the demands of the environment.

In terms of the challenge we provided to the RCs, we found three-bit functions to be relatively easy to approximate, while five-bit functions were more difficult, essentially requiring more resources. However, three-bit recursive functions were the most difficult, where we noted some recursive functions that are essentially impossible to approximate with this system, which is likely due to a long memory requirement.

Approximating the five-bit functions depends on a large reservoir and input size, and more strongly depends on $\bar{K}$. The five-bit function space is massive, so while we were only able to sample a tiny fraction of it, we clearly saw the effect of the dynamics that were close to criticality on approximation accuracy. As $\bar{K}$ increased toward chaotic dynamics, the approximation became very poor, leading to a large separation between dynamical regimes. Looking forward, it is likely that seven-bit or nine-bit function approximations would require increasingly more resources in terms of reservoir size, and have a greater dependence on dynamics. This seems to be the case for the median and parity functions, at least [50]. We found that the relative accuracy of reservoirs with $\bar{K}$ = 1, 2, or 3 could change, depending on the value of L, which is the input connectivity into the reservoir. This is most evident with the five-bit functions in the $\bar{K}$ = 2 accuracy plot, which showed a tapering off curve (convex), where improvement with reservoir size is not linear (Figure 6). In the curves, as L increases, we see the $\bar{K}$ = 1 and $\bar{K}$ = 2 accuracy curves crossing each other. One possible explanation for this is that the external perturbations to the reservoir’s nodes shift the dynamics of the reservoir to a less ordered regime, creating a higher fraction of perturbed nodes and resulting in a greater shift. So, the dynamics in $\bar{K}$ = 1 approach criticality with higher L values, while the dynamics in the $\bar{K}$ = 2 case moving past criticality, toward chaos, thus dropping in approximation accuracy. However, the effect of time-varying external perturbation has not been well studied for Boolean network dynamics. [50] showed that the mutual information between the reservoir and the signal increase monotonically as L increases for $\bar{K}$ = 1, so there may be other effects of increasing L at play. Although our results indicate a strong effect of mean in-degree ($\bar{K}$) in accuracy, it is possible that accuracy is also affected by other properties of reservoir topology; thus, additional in-degree distributions should be investigated.

We found that Boolean function sensitivity also plays a role in the accuracy, where more sensitive functions are more difficult to approximate. The sensitivity can be broken down further into the activities of the function variables. Functions appear to be more difficult to approximate if the activities of temporally distant variables are higher. For non-recursive functions, increasing N and L is sufficient to enable BN RCs to approximate sensitive functions; however, for non-recursive functions, large reservoirs with high values of are not as affected. This points to a large portion of the reservoir needing perturbation to avoid the problems that are related to function argument sensitivity.

However, for recursive functions, even large reservoirs are affected by Boolean function sensitivity. This is likely due to the non-smooth output, which is difficult to approximate. Further, the recursive sliding-window filtering formulation essentially creates infinite memory, albeit fading. Many questions remain, such as what is needed for good approximations of recursive functions, and whether there is a change to the model that might help. That said, for the most part, many recursive functions are approximated very well, and that may be enough.

Overall, having an idea of the sensitivity of the processing task will inform the minimum parameter values that are necessary for a successful reservoir. This is intriguing from a biological standpoint, and it is worth exploring how signal responses that are more sensitive, especially to more historic signal values, are handled. The inability of approximating certain functions may not impinge on the use for modeling biology. When viewed through the lens of ‘complex adaptive systems’, recursion is a key characteristic, and one that BN RCs would need to address in order to model biological systems.

## 5. Conclusions

Reservoir computers with adequately sized Boolean network reservoirs and input sizes show that even a fixed topology is flexible enough to approximate most Boolean functions, when applied to signals in a sliding-window fashion, with high accuracy. On one hand, BN RCs have structural similarity to biological systems such as gene regulatory networks, while on the other hand, they are useful objects for operating on signals, such as through the application of recursive filters to biological data. We observe that Boolean networks that are closer to the critical dynamics regime lead to more flexible reservoirs. Recursive functions are found to be the most difficult to approximate, owing to the necessity for approximating a function with hidden variables. Although most recursive functions can be approximated, we find that some can never be approximated. Boolean functions with more arguments are more strongly dependent on the dynamics of the system for accuracy, leading to a greater separation in the accuracy curves that correspond to different dynamical regimes. We find a connection between Boolean function sensitivity, an estimate of the smoothness of a function, and the ability to approximate a given function.

## Figures and Tables

**Figure 1 entropy-20-00954-f001:**
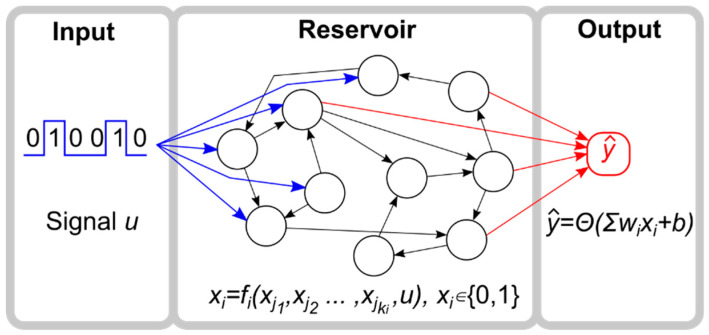
Reservoir computer layout. The reservoir computer (RC) is composed of a binary input node, a Boolean network reservoir, and a binary output node.

**Figure 2 entropy-20-00954-f002:**
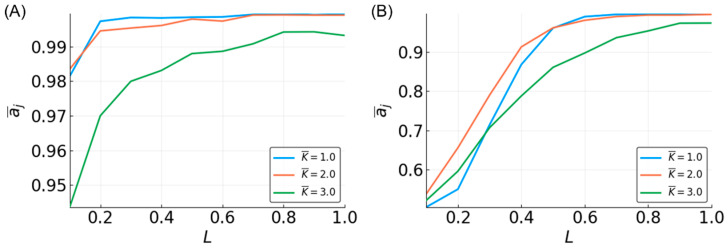
Mean accuracy, ${\bar{a}}_{j}$, vs. L for the three-bit median (**A**) and parity (**B**) functions for different $\bar{K}$-valued reservoirs with N = 500.

**Figure 3 entropy-20-00954-f003:**
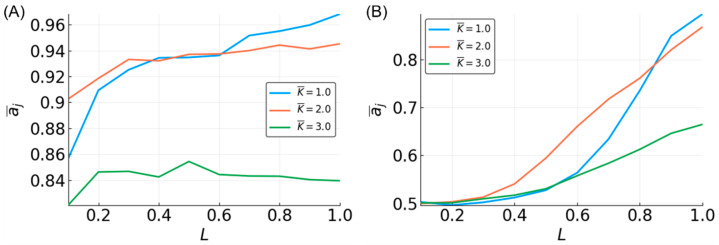
Mean accuracy, ${\bar{a}}_{j},$ vs. L for the five-bit median (**A**) and parity (**B**) functions for different $\bar{K}$-valued reservoirs with N = 500.

**Figure 4 entropy-20-00954-f004:**
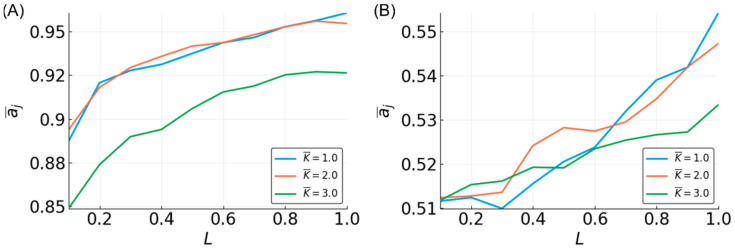
Mean accuracy, ${\bar{a}}_{j}$, vs. L for the recursive three-bit median (**A**) and parity (**B**) functions for different $\bar{K}$-valued reservoirs with N = 500.

**Figure 5 entropy-20-00954-f005:**
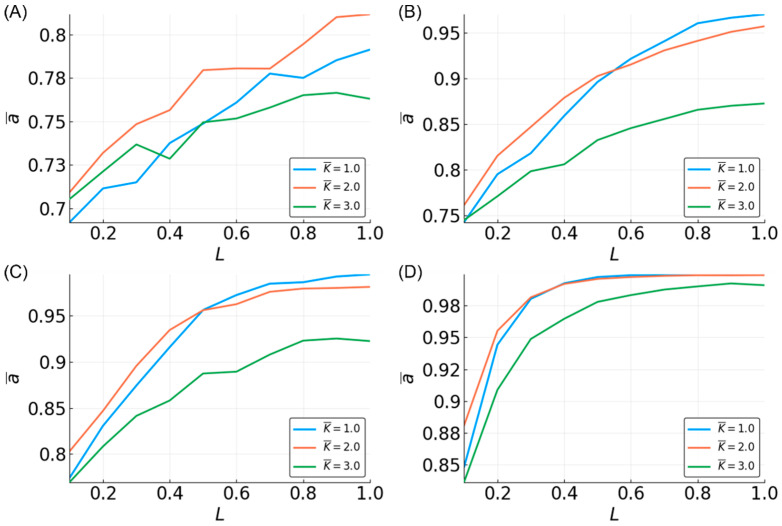
Mean accuracy, $\bar{a}$, vs. L for all of the three-bit functions for different $\bar{K}$-valued reservoirs. Different sizes of reservoirs are shown: N = 10 (**A**); N = 50 (**B**); N = 100 (**C**); and N = 500 (**D**).

**Figure 6 entropy-20-00954-f006:**
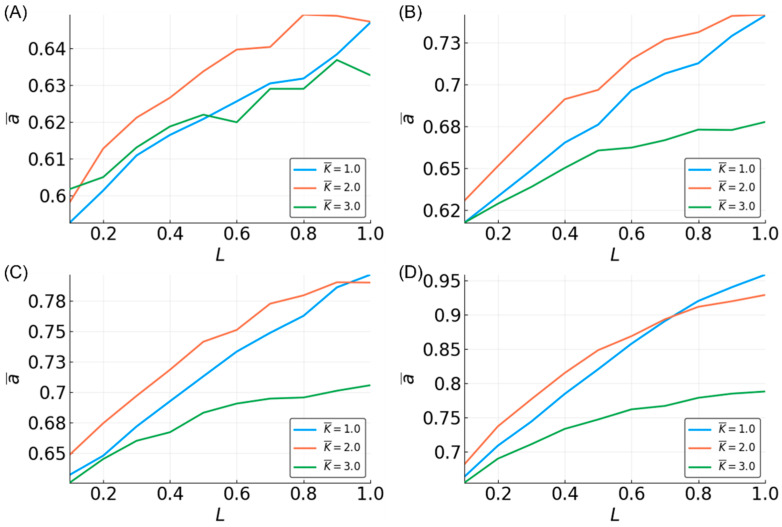
Mean accuracy, $\bar{a}$, vs. L for five-bit functions for different $\bar{K}$-valued reservoirs. Different sizes of reservoirs are shown: N = 10 (**A**); N = 50 (**B**); N = 100 (**C**); and N = 500 (**D**).

**Figure 7 entropy-20-00954-f007:**
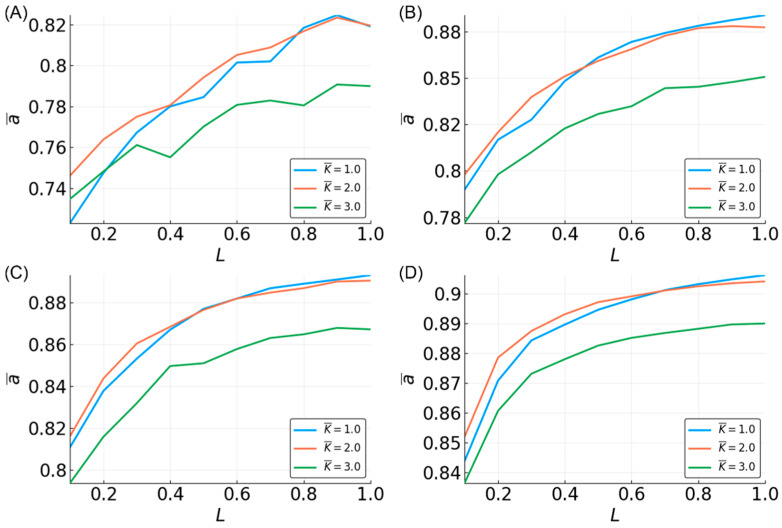
Mean accuracy, $\bar{a}$, vs. L for all of the recursive three-bit functions for different $\bar{K}$-valued reservoirs. Different sizes of reservoirs are shown: N = 10 (**A**); N = 50 (**B**); N = 100 (**C**) and N = 500 (**D**).

**Figure 8 entropy-20-00954-f008:**
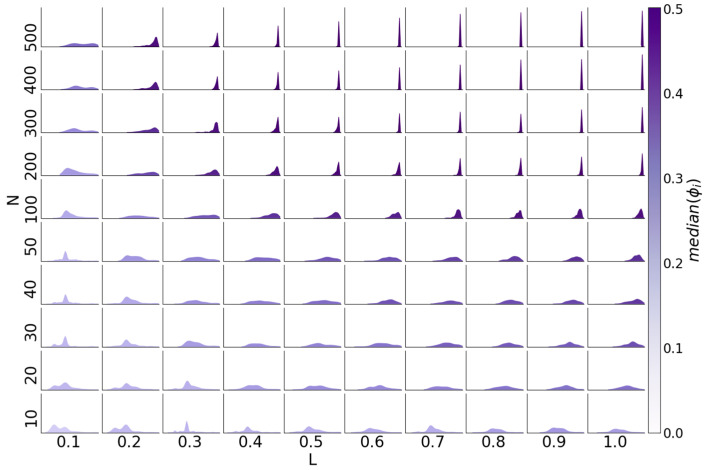
Histogram of $\Phi_{i}$ across all 100 reservoirs for each N, L with $\bar{K}$ = 2 for three-bit functions. Each subplot represents the density for all of the reservoirs with one N and L, with the x-axis being Φ, and the y-axis being the number of reservoirs [0, 256].

**Figure 9 entropy-20-00954-f009:**
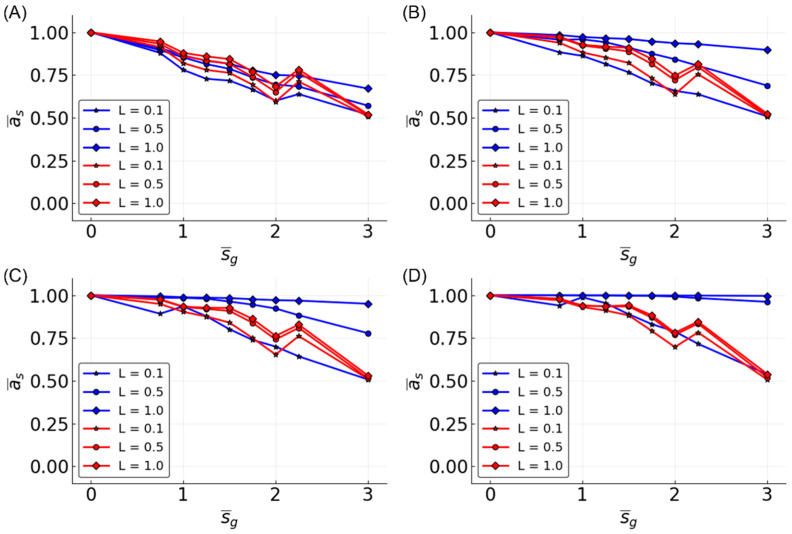
Mean accuracy, ${\bar{a}}_{s}$, vs. function average sensitivity, ${\bar{s}}_{g}$. Three-bit functions are shown in blue, and recursive three-bit functions shown in red with L = 10 (stars), 50 (circles), and 100 (diamonds). Four different values for N are shown: (**A**) N = 10; (**B**) N = 50; (**C**) N = 100; and (**D**) N = 500. Only reservoirs with $\bar{K}$ = 2 are shown. See Appendix A for $\bar{K}$ = 1 and $\bar{K}$ = 3 (Appendix A).

**Figure 10 entropy-20-00954-f010:**
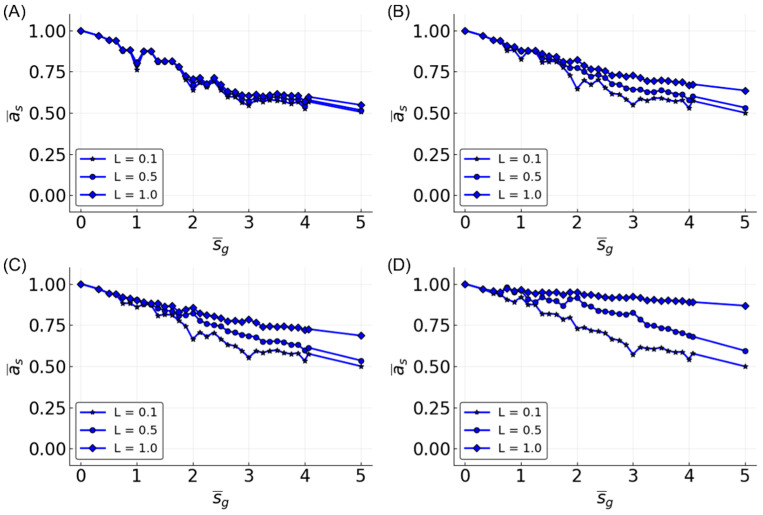
Mean accuracy, ${\bar{a}}_{s}$, vs. function average sensitivity, ${\bar{s}}_{g}$ for five-bit functions. L = 0.1 (stars), 0.5 (circles), and 1 (diamonds) are given in each plot. Four different values for N are shown: (**A**) N = 10; (**B**) N = 50; (**C**) N = 100; and (**D**) N = 500. Only reservoirs with $\bar{K}$ = 2 are shown.

**Figure 11 entropy-20-00954-f011:**
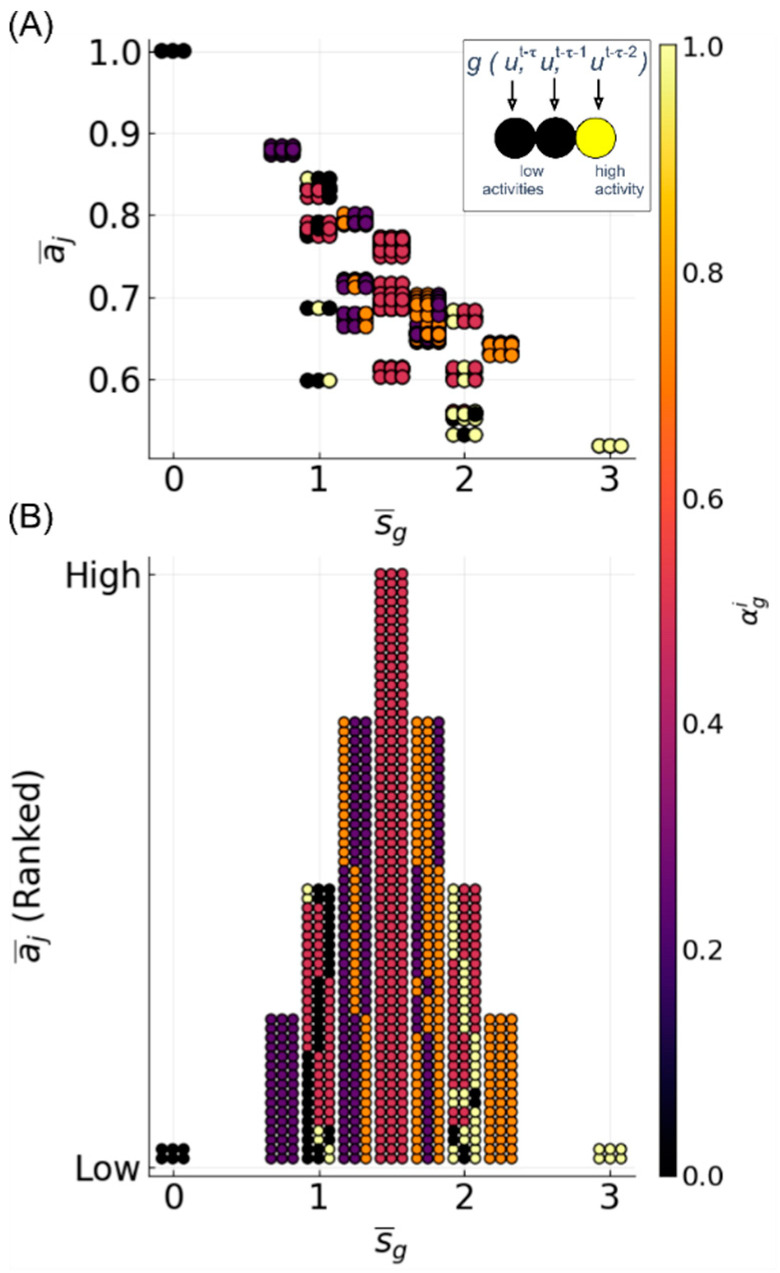
Example of mean function accuracy vs. average sensitivity with activities of each variable displayed. Data shown for three-bit functions: N = 10, L = 0.1, and $\bar{K}$ = 2. (**A**) Each of 256 functions is visualized as a horizontal triplet of circles, where each circle corresponds to a variable (left to right, $u^{t-\tau }$, $u^{t-\tau -1}$, $u^{t-\tau -2}$), as colored by its activity (inset). For example, the parity function can be seen at ${\bar{s}}_{g}$ = 3, ${\bar{a}}_{j}\approx$ 0.5, and $A_{g}$ = [1, 1, 1]. (**B**) In order to more clearly see the relationship between the distribution of activity and accuracy, functions are plotted by ranked accuracy rather than absolute accuracy. Here, the height of the columns is a result of the number of functions with a given ${\bar{s}}_{g}$, and does not reflect absolute accuracy.

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
