# Peer review of "Flexibility of Boolean Network Reservoir Computers in Approximating Arbitrary Recursive and Non-Recursive Binary Filters"

_entropy, 2018, doi:10.3390/e20120954_

Round 1

Reviewer 1 Report

In ”Flexibility of Boolean Network Reservoir Computers in Approximating Arbitrary Recursive and Non-recursive Binary Filters“, Echlin et al. evaluated RBNs with various configurations as reservoirs in echo state networks. 

In general, RBNs as reservoirs are considerable replacements for state of the art reservoirs using perceptrons, which was shown in recent research.

Main point:

The results of the manuscript would be improved by comparing the Boolean network reservoirs with standard neural (heterogeneous) reservoirs of the same size. Do Boolean network reservoirs outperform them with respect to flexibility and accuracy or reduce the computational complexity?

Minor issues: 

Spelling errors such as ”The node in-degree need to be constant“

Figure 10 should be revised. The legend can be places aside of (A) and (B) in order to prevent a separate new line.

Author Response

Reviewer 1: Thank you for your help!

Please see the uploaded pdf for our responses.

Reviewer 2 Report

The paper presents an interesting study on the use of Random Boolean Networks as reservoir in a Reservoir Computing system. To the best of my knowledge, this if the first thorough study on this subject.

The paper presents results produced by a wealth of experiments; the experimental analysis is sound and deep. Overall the paper is well structured, but some work is needed to make the reading smoother and to clarify some points. Comments follow to suggest some amendments in this direction.

* Outputs: I presume that the choice of considering all the N nodes of the network in the output function comes from previous work in RC and it may be also justified by the fact that a regression model is used with just N weights. Nevertheless, I think that this point should be clearly stated. Moreover, a couple of questions should be addressed:
1) are the final weights connected to input nodes somehow different wrt the weights of the other nodes of the network?
2) what is the overall distribution of the final weights?

* Initialization: are the initial values of the non-input nodes set randomly? Does the initialisation scheme affect the final performance?

* Average sensitivity vs accuracy: in line 456 it is stressed that at specific values of s_g the trend deviates from linearity. There is probably something I'm missing, but it is not clear to me whether the subsequent study considering also the activity of variables explains also the fact that these peculiar s_g values are integer.

* The introduction starts with the description of fruit flies olfactory system and then abruptly changes to a general description of nonlinear dynamical systems as information processing systems. I usually appreciate the incipit of a paper in terms of a prologue, but this should be emphasised. I then suggest either to clearly separate the first paragraph from the actual introduction or to connect less abruptly the first paragraph to the second.

* Minor remarks:

- line 68: This can be [???] through
- line 194: here I'd suggest to remind that also median and parity functions will be used
- line 207: y was previously used to denote a scalar boolean value, whilst here it is clearly referring to a vector. I suggest to disambiguate the notation
- line 346: K=2 [DEL is] decreases
- line 391 and 395: a blank space is missing after L
- line 400: right bracket missing
- line 497: frankly, the plot in Fig. 11A is quite puzzling and requires time to be understood. Would it be possible to produce a different and clearer graphic explanation of the same information?
- line 561: "what is be needed for". Do you mean "what is needed"?

Author Response

Thank you for the great comments!  Please see the uploaded pdf for our responses.

Reviewer 3 Report

The authors studied random Boolean Networks as reservoir computers (BNRCs). The learning tasks used are the approximation of 3-bit and 5-bit non-recursive functions and 3-bit recursive binary functions. Performance of BNRCs was investigated extensively by varying network size, average in-degree, and the fraction of nodes directly receiving inputs. In particular, the dependence of flexibility of BNRCs on these parameters and influence of sensitivity of binary functions to approximation accuracy were explored in detail. 

It seems that the results obtained in this paper extend and compensate those of previous work (Refs. 38, 50). The paper is organized clearly and is readable. It contains several useful insights for using BNs as reservoirs. In these regards, publication of this paper has merit to researchers working on BNs and RCs. However, the following comments should be considered before publication. 

List of Comments

==========

Page 1, L27--32:

Starting a paper on RCs from the description of the olfactory system of fruit flies is curious. This paragraph seems to be an example of the sentence at the beginning of the next paragraph. Please rewrite these two paragraphs appropriately. 

Page 5, L199--201:

How did you obtain the 150 random binary sequences? From different initial conditions? Is there any transient period before sampling the sequences? Please clarify these points. 

Page 7, L251:

It seems that the equation is incorrect. I guess that d_t is 1 if the condition on the right-hand-side is satisfied, and d_t is 0 otherwise. 

Page 8, L302--303:

I guess that the sentence means that the authors used networks with constant in-degrees while Synder et al. used networks with Poisson degree distributions. Please make this explicit in the main text. 

Page 13, L383:

Please give more words for justifying Phi_i as a measure of flexibility. My concern is that mad(a_{ij}) can be arbitrarily small in principle. This implies that Phi_i may diverge. So, even when a_{ij}s are close to 0.5, if mad(a_{ij}) is sufficiently small, then Phi_i can take a large value. On the other hand, it seems that the x-axis of Fig. 8 has a finite range. Please resolve this concern. 

Page 13, L396:

It seems that there is no clearly defined ``transition point''. Please replace this expression with a suitable one. 

Page 14, L431:

``)'' on the left of the symbol of addition modulo 2 should be placed on the right of e_i. 

Page 15, L476--477:

Please provide reasons for the sentence ``It is possible ...''. If this is one of future work, please state it explicitly. 

Supplementary Materials:

In Figures S1-S3, please explicitly write the correspondence between panels and the values of N. 

==========

Author Response

Thank you for your help!  Please see the uploaded pdf for our responses.
